# Reflection System for the Abstraction and Reasoning Corpus

## Abstract

The Abstraction and Reasoning Corpus (ARC) benchmarks broad generalization in artificial intelligence, and presents a significant challenge to existing machine learning models and program synthesis solvers. In this work, we introduce a Reflection System for ARC. It combines Large Language Models (LLMs) and a program synthesis solver based on a Domain Specific Language (DSL). We analyse the accuracy of LLMs on ARC and demonstrate unsatisfactory results. We create AugARC, an augmented ARC benchmark, which consistently improves the performance of LLMs compared to the normal ARC benchmark. Using augmented ARC data, we fine-tune LLMs and observe a significant gain in ARC accuracy after training. By utilizing reflection, we combine LLMs and a previous DSL solver into our Reflection System for abstraction and reasoning. Our approach outperforms the previous publicly available ARC systems that consist solely of LLMs or DSL solvers. The proposed Reflection System motivates research to advance previous ARC attempts by combining the advantages of LLMs and program synthesis solvers with reflection.

## Introduction

Incorporating abstract reasoning into machines has been an active research topic since the 1955 Dartmouth AI workshop (McCarthy et al. 2006). Despite the significant progress in machine learning, today's AI systems still lack human-level abstract reasoning (Korteling et al. 2021). Studies have shown that digital systems are significantly inferior to humans in terms of abstract cognitive abilities (Boden et al. 2017; Shneiderman 2020).

To address the gap between human intelligence and AI models, François Chollet created the Abstraction and Reasoning Corpus (ARC) (Chollet 2019). ARC consists of 1000 visual tasks, that capture essential aspects of abstraction and analogy. The ARC tasks are split into 400 for training, 400 for evaluation and 200 hidden tasks for testing. A Program Synthesis approach from 2020 solved 40% of the complete evaluation set (Icecuber 2023), and a voting ensemble from 2024 achieved 40.25% (Bober-Irizar and Banerjee 2024).

State-of-the-art systems that attempt to solve the ARC test set use heuristic search. Such models are heavily handcrafted and designed entirely with the goal of solving ARC. Recent attempts have tried solving ARC with Graph Abstractions (Xu, Khalil, and Sanner 2023) and Generalized

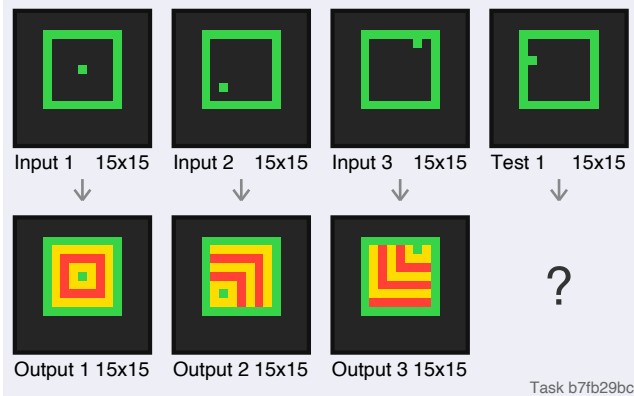

Figure 1: Visualisation of an ARC task. The test-taker is provided with some input-output pairs as examples. The objective is to recognise the transformation used in the given input-output pairs and apply it to the test input grid to obtain the test output grid.

Planning (Lei, Lipovetzky, and Ehinger 2024). However, these two approaches have only been tested on a subset of ARC evaluation data. Some attempts have been made to use Large Language Models (LLMs) to solve ARC (Xu et al. 2023; Min 2023; Mitchell, Palmarini, and Moskvichev 2023), with some of the previous publications testing LLMs on the ARC evaluation set (Bober-Irizar and Banerjee 2024; Opiełka et al. 2024; Gendron et al. 2023; Lee et al. 2024b). Nevertheless, previous studies test LLMs only on subsets of the ARC evaluation data and do not attempt to build more advanced systems with reflection based on several LLMs.

Hence, we aim to fully explore the abilities of base LLMs on ARC and how those can be combined in systems with reflection. We introduce a new augmented ARC (AugARC) benchmark tailored towards LLMs, which shows consistently improved performance across all tested LLMs compared to the normal ARC. We show the benefit of fine-tuning

LLMs on augmented ARC data. Finally, we build a Reflection System which solves 5 more evaluation tasks than a previous system that combines multiple ARC solvers (Bober-Irizar and Banerjee 2024).

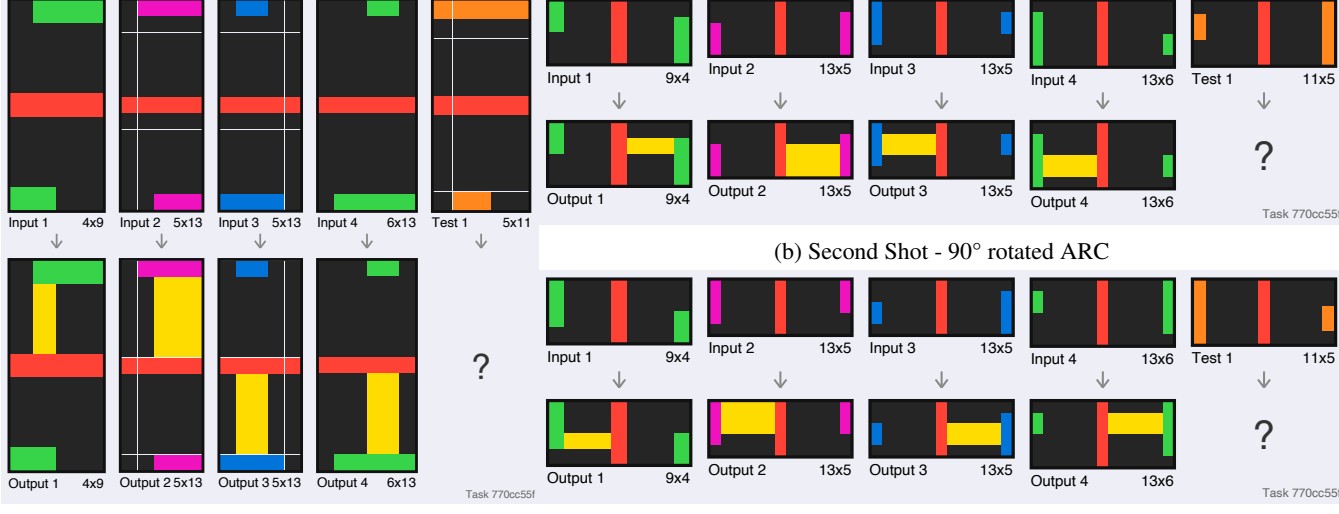

Figure 2: Evaluation Task of the 3-shot AugARC Benchmark. The first shot is a normal ARC evaluation task, while the second and third shots are 90° and 270° rotated. All three shots are represented as a 2D matrices of numbers, each one representing a different colour. The figure showcases the three shots in as coloured grids for demonstration purposes.

# AugARC: Augmented ARC for LLMs

ARC training data can be utilised to fine-tune LLMs and improve their performance in evaluation and test sets. One potential issue with this approach is the size of the training set - it contains only 400 samples. Since LLMs have billions of parameters, they usually cannot be effectively trained on smaller datasets and instead require more samples. Therefore, due to its small size, the ARC training dataset limits the ability to fine-tune LLMs for improved broad generalization and reasoning.

## Augmented Training Data

To overcome the limited number of ARC training tasks, we implement an augmentation procedure that can significantly extend the training dataset. Our approach expands the ARC training set by applying the following transformations:

- **Rotation:** clockwise rotation of each ARC grid for a given task by 90° or 270°.
- **Flipping:** flips each ARC grid of a task horizontally (along the y-axis) and vertically (along the x-axis).
- **Permutations:** rearranges the sequence of demonstration input-output pairs before the test input grid. We set a threshold for the maximum number of permutations per task to produce datasets of various sizes.

Depending on the transformations applied and the maximum number of permutations applied, the augmented ARC training datasets vary from 2000 up to over 18 million tasks.

## 3-Shot AugARC Benchmark

A key reason for the relatively scarce ARC research on LLMs is the lack of a textual version of the benchmark.

| Dataset Size | Max Permutations |
|---|---|
| 2 000 tasks | - |
| 4 000 tasks | 2 |
| 5 715 tasks | 3 |
| 7 430 tasks | 4 |
| 9 145 tasks | 5 |
| 18 668 610 tasks | All |

Table 1: Size of the augmented ARC training datasets according to the maximum number of permutations. All datasets include 90° and 270° rotations, and horizontal and vertical flipping. The augmented datasets range in size from 2000 to 18 million tasks.

The only benchmark suitable for LLMs that resembles Chollet's visual ARC (Chollet 2019) is the AI2 Reasoning Challenge (Clark et al. 2018; Pătraş et al. 2022). AI2 is a multi-choice question answering benchmark that focuses on assessing reasoning. Although AI2 is a more popular and well-established reasoning benchmark for LLMs compared to Chollet's ARC (Chollet 2019), the latter is more effective at evaluating broad generalization abilities due to its hand-crafted abstract logic.

Identifying that the lack of a textual ARC benchmark is a significant barrier for evaluating LLMs, we create the AugARC Benchmark. The AugARC Benchmark provides an easy and unified way to evaluate LLMs on 3-shot accuracy on reasoning tasks. In AugARC, each ARC task starts with a textual description explaining the format of the problem. Each ARC grid is represented as a 2D matrix of numbers.

**AugARC Input to LLMs** The first prediction is based on a normal ARC task, whereas the second and the third ones

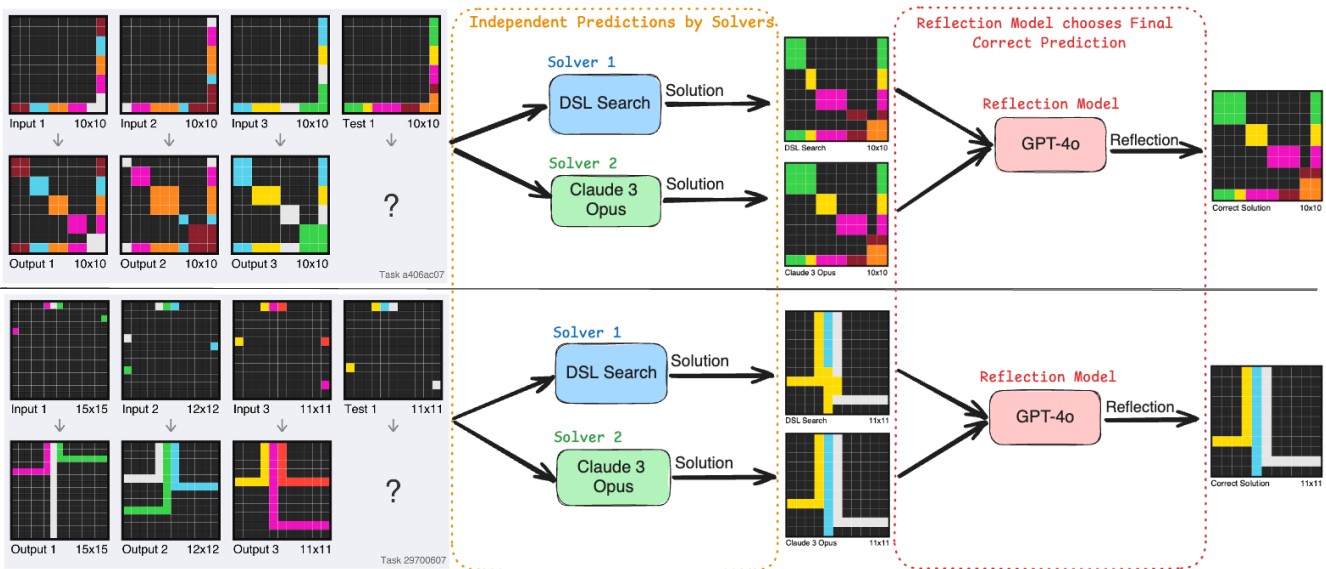

Figure 3: Reflection System - execution on two ARC evaluation tasks. Initially, multiple solvers make independent predictions on the task. Then, the task and the prediction are presented to the reflection model, which chooses the correct final prediction. In the example, solver 1 is based on program synthesis (DSL Search) and solver 2 is an LLM (Claude 3 Opus). The reflection model is an LLM (GPT-4o). Both task flows are actual demonstrations of how the Reflection System performs on ARC evaluation tasks. In both cases, the Reflection System produces a correct final solution.

are 90° and 270° clockwise rotated versions of the same task. The AugARC Benchmark is tailored for LLMs, as these models process inputs in an auto-regressive, sequential manner. By rotating the ARC tasks, LLMs are presented with a different sequence of numbers (2D matrices) which contain the same abstract logic.

**Reproducing ARC Solutions from AugARC Outputs** Although the second and third shot in AugARC are based on rotated ARC tasks, the output of the LLMs can easily be transformed back to a solution to the original ARC problem. Once an output is generated by the LLM, it is simply rotated back in an anticlockwise direction. In this way, AugARC only changes the input representation of the ARC problems: the outputs by the models are then rotated to valid ARC solutions. This process ensures that the results with the AugARC approach are directly comparable to previous ARC attempts.

## Fine-tuning LLMs on ARC tasks

Although LLMs have shown impressive capabilities, they can sometimes hallucinate and are therefore regarded as unreliable in reasoning tasks. One potential way to reduce such hallucinations and improve performance on abstract logical tasks is to fine-tune LLMs. Due to the limited size of the ARC training dataset (400 tasks), previous studies have not attempted to train LLMs on ARC. Our proposed augmentation of ARC allows us to overcome this limitation and have sufficient ARC data to fine-tune LLMs.

For efficient training of LLMs, we use Quantized Low-Rank Adaptation (QLoRA) with 4-bit NormalFloat (NF4) quantization (Dettmers et al. 2024). Low-Rank Adaptation constrains the update of a pre-trained weight matrix

$W_0 \in \mathbb{R}^{d \times k}$ with a low-rank decomposition $W_0 + \Delta W = W_0 + BA$, where $B \in \mathbb{R}^{d \times r}$, $A \in \mathbb{R}^{r \times k}$, and the rank $r \ll \min(d, k)$ (Hu et al. 2021). During training, $W_0$ is frozen and does not receive gradient updates, while $A$ and $B$ contain trainable parameters. Both $W_0$ and $\Delta W = BA$ are multiplied with the same input, and their respective output vectors are summed coordinate-wise (Hu et al. 2021).

Using QLoRA, we fine-tune LLMs on an augmented ARC training dataset consisting of 2000 tasks 1. Due to a significant increase in computational complexity, we avoid fine-tuning the models on some of the bigger augmented ARC training sets from Table 1. For the same reason, we only train LLMs with parameters ranging from 7 to 13 billion.

## Reflection System

A previous promising approach which solves 40.25% of the ARC evaluation tasks combines solutions from different ARC solvers (Bober-Irizar and Banerjee 2024). This approach utilizes a voting ensemble of systems that "votes" for the predictions of an LLM, a Program Synthesis solver and a Neuro-symbolic model (Bober-Irizar and Banerjee 2024). The voting ensemble outperforms systems that are solely based either on LLMs or Program Synthesis such as the DSL Search (Icecuber 2023).

The encouraging result of the voting ensemble motivates further research into combining different architectures into a complex system for enhanced ARC performance. Although the voting ensemble achieves promising ARC accuracy, it lacks any "intelligent" analysis of potential solutions and instead uses a weighting algorithm (Bober-Irizar and Banerjee

2024). In order to build upon this limitation and combine multiple previous attempts into a new complete approach, we propose a Reflection System for ARC.

Our approach relies on solvers that could have various architectures - for example, LLMs or Domain Specific Languages using Program Synthesis. When predicting the correct solution to an ARC task, the Reflection System executes in two main stages, as visualised in Figure 3.

### Predictions in the Reflection System

In the first stage, each solver makes a prediction on the given ARC task. Each solver solver independently and cannot access the outputs of other solvers. Once a solver has produced a prediction for the ARC task, it passes the solution to the reflection model.

### Reflection over all Prediction

The second stage of our approach is inspired by previous studies on self-reflection (Lee et al. 2024a; Renze and Guven 2024), in which LLMs refine their responses based on feedback against previous outputs and, in this way, achieve more accurate predictions. In our system, the reflection model processes all generated predictions from all the ARC solvers. Conditioning on the given ARC task, the reflection model chooses the prediction from the solver that is most likely to be correct.

### Flexibility of the Reflection System

In the Reflection System, an ARC solver can be any model including LLMs, program synthesis approaches or neuro-symbolic models. Any number of solvers can be used, as the reflection model can easily process the outputs of various solvers. This makes our approach customisable, as each of its components - the ARC solvers and the reflection model, can easily be changed. This architectural design allows the Reflection System to be easily tested with various ARC solvers for finding the optimal ARC configuration.

## Experiments

We perform all experiments on the ARC evaluation set which consist of 400 tasks. By design, the ARC evaluation set is significantly more challenging than the training set (Chollet 2019). The creator of ARC, François Chollet, emphasised that the performance of intelligent systems should be measured by the fraction of tasks solved on the evaluation set (Chollet 2019). Therefore, we perform our experiments on the evaluation set and use 3 shots per task, as set out in the ARC design (Chollet 2019).

To present fully reproducible results, all experiments are executed on the complete evaluation set. Some previous solvers have been evaluated on a subset of the ARC evaluation data, making it difficult to understand the true performance of the solver. Our testing approach ensures that future studies could easily use our results for direct comparison with new ARC solvers.

### Performance on base ARC and AugARC

We start our experiments with LLMs on the base ARC benchmark, shown in Table 2. The ARC accuracy across 7-13 billion models ranges from 5 to 9 correctly solved ARC evaluation tasks. Bigger LLMs solve slightly more ARC tasks, from 7 to 20 solved tasks, with Gemini Pro achieving the highest accuracy (20).

| Model | ARC | AugARC | Increase |
|---|---|---|---|
| Llama-2 7B | 5/400 | 7/400 | 29% |
| Mistral 7B | 9/400 | 15/400 | 67% |
| Llama-2 13B | 5/400 | 8/400 | 100% |
| Llama-2 70B | 7/400 | 14/400 | 100% |
| Mixtral 8x7B | 9/400 | 18/400 | **125**% |
| Gemini Pro | 20/400 | **33/400** | 65% |

Table 2: Performance of LLMs on ARC and AugARC (on the evaluation set). There is a consistent increase of the accuracy of LLMs when using the AugARC inputs compared to using the base ARC ones (29-125%).

Using the same LLMs, we evaluate the performance on AugARC. For all LLMs, there is a clear accuracy improvement on AugARC compared to base ARC. The increase varies from 29% for Llama-2 7B up to 125% for Mixtral 8x7B, with the majority of models achieving at least 60%.

The significant improvement in all LLMs on AugARC compared to ARC suggests that changing the grid structure of the tasks for the second and third shots leads to increased accuracy. LLMs process the ARC tasks sequentially, and thus are directly influenced by the exact order of the grids. Based on the results, we conclude that the proposed AugARC benchmark is well suited for LLMs.

Since AugARC results are directly comparable to ARC, we proceed to use AugARC for the remainder of our experiments.

### ARC accuracy across LLMs

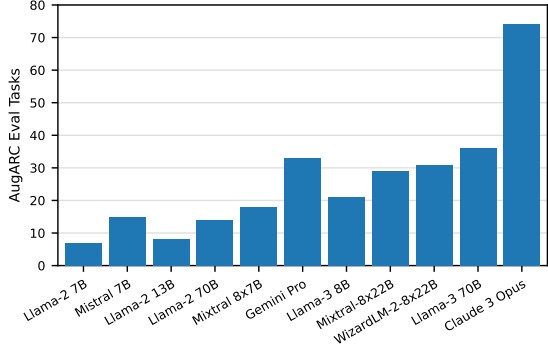

Figure 4: ARC evaluation tasks solved by LLMs. Claude 3 Opus solves the most ARC tasks (74).

The ARC accuracy of LLMs ranges between 7 up to 74 solved tasks, as visualised in Figure 4. The best performance

of a small model is achieved by Llama-3 8B (21). Some bigger open-source LLMs can solve more than 30 ARC tasks, with Llama-3 70B achieving 36. The highest number of solved ARC task, 74, is by Claude 3 Opus.

The ARC results demonstrated some variability in performance across LLMs. Bigger models appear to be more accurate on ARC compared to smaller ones. Most LLMs achieve an accuracy in the range of 10-35 tasks, with the only exception being Claude 3 Opus with 74.

## Performance of Fine-tuned LLMs on ARC

To observe whether we can reduce the performance gap between smaller and bigger LLMs on ARC, we fine-tune the 7 and 13B models. The models are fine-tuned on the training set only using a single Nvidia A100 80GB GPU.

The results in Table 3 show that the fine-tuned LLMs solve between 18 and 34 ARC evaluation tasks. Training benefited all the models substantially - the small fine-tuned Llama-2 7B and 13B achieved a performance on par with the base versions of the significantly bigger models such as Llama-2 70B. After fine-tuning, Mistral 7B outperforms the standard Mixtral 8x7B by 5 correct tasks. The highest result of 34 correct solutions after fine-tuning by Llama-3 8B is impressive, as it outperforms Gemini Pro.

| Model | Base | Fine-tuned | Increase |
|---|---|---|---|
| Llama-2 7B | 7/400 | 21/400 | **200%** |
| Mistral 7B | 15/400 | 23/400 | 53% |
| Llama-2 13B | 8/400 | 18/400 | 125% |
| Llama-3 8B | 21/400 | **34/400** | 62% |

Table 3: Results of base and fine-tuned LLMs on the ARC evaluation set. The increase column shows the improvement in accuracy from a base LLM compared to its fine-tuned version. All LLMs consistently show improved ARC performance after fine-tuning, ranging from 62% to 200%.

The results in Table 3 demonstrate a significant increase in ARC performance across all fine-tuned LLMs compared to their base versions. The accuracy improvement after training varies between 53% in Mistral 7B up to 200% in Llama-2 7B. While Llama-2 7B and 13B both achieve more than 100% improvement - 125% and 200% respectively, Mistral 7B and Llama-3 8B improved in the range of 50% to 65%.

Our results suggest that training small LLMs on the AugARC dataset consistently improves their performance. In particular, fine-tuning smaller LLMs (7-13B parameters) is so effective that it can lead to better ARC performance than significantly larger base LLMs.

## Solution Overlap and Gain Measure

To motivate our reflection approach to ARC, we show the benefit of combining ARC solutions from base and fine-tuned LLMs together with Program Synthesis solvers.

The ratio of overlapping solutions between different ARC solvers is visualised in Figure 5a. The numbers in Figure 5a refer to the proportion of overlapping tasks solved by the systems on the left and on the bottom. For each pair

of LLMs, there is an overlap between 0.5 and 0.9 in their correct ARC solutions. A lower overlap can be observed between the base LLMs and the fine-tuned ones. For example, a fine-tuned Mistral 7B has an overlap of only 0.52 with standard models such as Mixtral 8x22B and Llama-3 70B. The low overlap in the solutions between fine-tuned and base LLMs indicates that training the models leads to correct solutions to new ARC tasks, which have previously not been solved by the base LLMs. The solution overlap between LLMs and a Program Synthesis solver, DSL Search (Icecuber 2023), ranges between 0.69 and 0.9.

We also measure the gain from adding a second system when testing the models on ARC. Figure 5b shows how much the base systems on the left would benefit from adding the solutions from the models at the bottom. We visualise how much the base systems can gain from utilizing new correct solutions from the second systems. For most LLMs, the gain of adding the solutions from another LLM is between 3 and 24. The gain between every two LLMs is slightly skewed by Claude 3 Opus due to its substantially better performance than any other LLM.

Most LLMs only solve 3 to 6 new tasks compared to the DSL Search (Icecuber 2023). Importantly, Claude 3 Opus could contribute 23 new correct solutions to the DSL Search, leading to a substantial improvement in ARC accuracy. This encouraging result motivates a new approach, which can effectively combine solutions from LLMs such as Claude 3 Opus with Program Synthesis solvers such as DSL Search (Icecuber 2023).

## Performance of the Reflection System

We experiment with different Reflection System configurations based on two or three ARC solvers and with different reflection models. Since the current best publicly available ARC performance is by a Program Synthesis solver (DSL Search (Icecuber 2023)), we always include it as a solver in all of our reflection system experiments. We also always include the LLM with highest ARC accuracy as a solver (Claude 3 Opus). We experiment with base and fine-tuned LLMs for the reflection models and a potential third solver to find the reflection system configurations which achieve the highest ARC accuracy.

Table 4 shows that the ARC performance by different reflection system configurations varies between 133 and 166 solved evaluation tasks. In a 2-solver setting, with DSL Search and Claude 3 Opus, Llama-3 70B struggles as a reflection model, solving only 133 tasks. GPT-4-turbo and GPT-4o perform significantly better as reflection models, solving 165 and 166 ARC tasks respectively. When adding a fine-tuned Llama-3 8B as a third solver, the reflection system solves 163 ARC tasks.

Our best 2-solver and 3-solver configurations both outperform the best single LLM, Claude 3 Opus (74), and the best Program Synthesis approach, DSL Search (160). Based on the results, we argue that the proposed Reflection System is an effective approach for combining LLMs and Program Synthesis solvers for enhanced ARC performance.

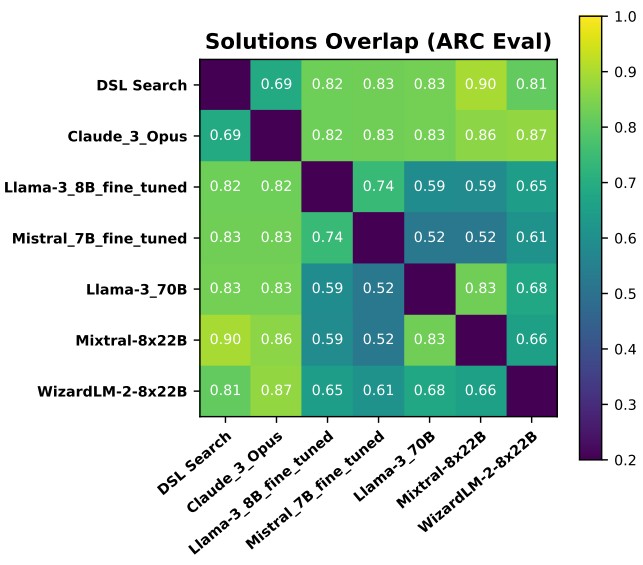 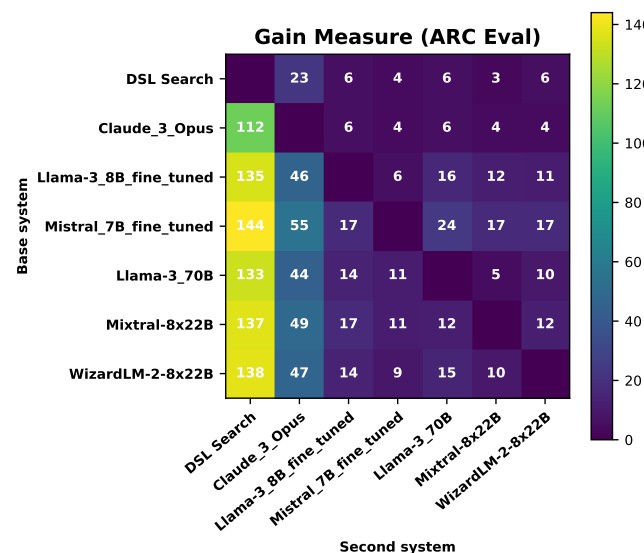

(a) Overlapping ARC tasks between the system on the left and the one on the bottom.

(b) Number of new solutions from adding a second system (bottom) to the first one (left).

Figure 5: Overlap of solutions and gain measure between systems. The systems are ordered by how much gain they add. In (a), the overlap ranges from 0.5 up to 0.9, with lower values between fine-tuned and base LLMs. In (b), Claude 3 Opus could add 23 new solutions to the DSL Search, while the remaining LLMs could add between 3 and 6.

| Solver 1 | Solver 2 | Solver 3 | Reflection Model | ARC Correct |
|----------|----------|----------|------------------|-------------|
| DSL Search | Claude 3 Opus | - | Llama-3 70B | 133/400 |
| DSL Search | Claude 3 Opus | - | GPT-4-turbo | 165/400 |
| DSL Search | Claude 3 Opus | - | GPT-4o | **166/400** |
| DSL Search | Claude 3 Opus | Fine-Tuned Llama-3 8B | Claude 3.5 Sonnet | 163/400 |

Table 4: Correctly solved ARC evaluation tasks in a 3-shot setting by different reflection system configurations. The best 2-solver model performance is with DSL Search and Claude 3 Opus as solvers and GPT-4o as a reflection model (166). A fine-tuned Llama-3 8B model achieves the top performance among 3-solvers (163).

## Previous Approaches

To demonstrate the effectiveness of our Reflection System, we compare the ARC performance of our optimal configuration to previous publicly available approaches. We present systems that have been tested on the complete ARC evaluation dataset and split the categories into LLMs, Neuro-symbolic models and Program Synthesis solvers. We also compare to a previous attempt combining different ARC solvers with a voting ensemble (Bober-Irizar and Banerjee 2024).

Table 5 shows that our approach outperforms all previous publicly available system types on the ARC evalua-

tion dataset. The Reflection System achieves a substantially higher accuracy than any previous LLM or neuro-symbolic model. Furthermore, it

| System Type | Method | ARC Correct |
|-------------|--------|-------------|
| Neuro-Symbolic | DreamCoder (Bober-Irizar and Banerjee 2024) | 18/400 |
| | CodeIt (Butt et al. 2024) | 59/400 |
| LLM | GPT-4 (Bober-Irizar and Banerjee 2024) | 32/400 |
| | Fine-Tuned Llama-3 8B | 34/400 |
| | Llama-3 70B | 36/400 |
| | Claude 3 Opus | 74/400 |
| Program Synthesis | Brute Force (Ainooson et al. 2023a) | 26/400 |
| | Neurodiversity solver (Ainooson et al. 2023b) | 45/400 |
| | DSL Search (Icecuber 2023) | 160/400 |
| Ensemble | Voting (Bober-Irizar and Banerjee 2024) | 161/400 |
| **Multiple Solvers** | **Reflection System** (Solvers: DSL Search, Claude 3 Opus; Reflection: GPT-4o) | **166/400** |

Table 5: Number of correctly solved ARC evaluation tasks across different system types. The highest ARC accuracy is achieved by the Reflection System (166).

outperforms the best available Program Synthesis solver (DSL Search), by 6 ARC tasks. the Reflection System also surpasses the accuracy of the voting ensemble (Bober-Irizar and Banerjee 2024), solving 5 more ARC tasks.

The results in Table 5 show that the Reflection System achieves a new best performance compared to all previous publicly available systems that have been tested on the complete ARC evaluation dataset.

# Related Work

Most of the previous ARC attempts can be split into two categories: Program Synthesis solvers and methods that rely on machine learning. The most successful publicly available attempt relies on efficient search implementations (Icecuber 2023). In contrast, machine learning implementations vary from neuro-symbolic models to the latest LLMs.

## Program Synthesis Solvers

A popular Program Synthesis solver for ARC is the DSL Search implementation by IceCuber which achieves 40% accuracy on the complete ARC evaluation dataset. The DSL solution is based on brute-force search. It applies transformations of varying depth in parallel and greedily stacking them to fit training samples (Icecuber 2023). The final prediction is ensembled based on the most solved training samples and least depth.

Another promising Program Synthesis approach is the the Generalized Planning for Abstract Reasoning (GPAR) solver (Lei, Lipovetzky, and Ehinger 2024). It casts an ARC problem as a generalized planning (GP) problem, where a solution is formalized as a planning program with pointers (Lei, Lipovetzky, and Ehinger 2024). On 160 of the 400 ARC evaluation tasks, GPAR outperforms the DSL Search by 10% (Lei, Lipovetzky, and Ehinger 2024). Nevertheless, due to the lack of GPAR results on the complete ARC evaluation set, in this work, we refer to the DSL Search as the current best Program Synthesis solver.

## Neuro-symbolic Models

Neuro-symbolic models have emerged as promising AI systems that aim at integrating the ability to learn from experience, and the ability to reason from what has been learned (Garcez et al. 2019). In neuro-symbolic computing, knowledge is represented in symbolic form, whereas learning and reasoning are computed by a neural network (Garcez et al. 2019).

The first Neuro-symbolic approach to solving ARC is DreamCoder (Alford 2021). It uses neural networks to guide its ability to write programs (Bober-Irizar and Banerjee 2024). An initial implementation of DreamCoder (Alford 2021) solves 2 ARC evaluation tasks, and an updated version with a Perceptual Abstraction & Reasoning Language (PeARL) achieves 18 (Bober-Irizar and Banerjee 2024).

Code Iteration (CodeIt) is a recent neuro-symbolic model that approaches ARC (Butt et al. 2024) as a programming-by-examples problem by training a policy to produce programs when shown demonstration examples (Butt et al. 2024). Experiments on the complete ARC evaluation set show that CodeIt solves 59 tasks, significantly outperforming previous neuro-symbolic approaches.

## Large Language Models

Previous research that explores LLMs on ARC has been primarily focused on OpenAI's GPT models (Mitchell, Palmarini, and Moskvichev 2023; Mirchandani et al. 2023; Moskvichev, Odouard, and Mitchell 2023).

Complete experiments on the 400 ARC evaluation tasks show that the best overall LLM is GPT-4 with 32 correct tasks (Bober-Irizar and Banerjee 2024), while the best open-source LLM is LLaMa-65B with 13 correct solutions (Bober-Irizar and Banerjee 2024). A general conclusion from existing work is that LLMs fail on simple ARC tasks (Xu et al. 2023).

## Ensembling Different System Types

A promising approach to ARC is to use a voting ensemble of systems: each system can propose an ARC solution which they "vote" for, added to a priority queue (Bober-Irizar and Banerjee 2024). Using this voting ensemble to combine the DSL Search (Icecuber 2023), DreamCoder and GPT-4 solutions achieves 161 correct tasks on the ARC evaluation set (Bober-Irizar and Banerjee 2024).

# Limitations

Since we did not have access to the data used for pre-training the LLMs, we cannot exclude the hypothesis that some models might have been pre-trained either on ARC tasks or on other very similar abstract problems. It can be argued that the significant improvement after fine-tuning demonstrates that most of the tested LLMs have not been pre-trained on ARC. Nevertheless, the substantially higher ARC results by Claude 3 Opus compared to all other LLMs raise some concerns that this model might have been pre-trained on ARC.

Additionally, our reflection approach lacked communication and collaboration between the solvers. The independence between the solvers in the Reflection System can limit its flexibility. Another potential limitation is that most of the correct solutions generated by our approach are produced by the DSL search (160 out of 166).

# Conclusion

We proposed a Reflection System, which effectively combines ARC solutions from Large Language Models and a Program Synthesis solver. We demonstrate that the Reflection System can easily be configured to work with a different number of solvers and various reflection models. On the complete ARC evaluation dataset, the Reflection System with 2 and 3 solvers outperforms previous approaches such as LLMs, Neuro-symbolic models, Program Synthesis, and a voting ensemble.

In future work, the Reflection System can be extended with more than 3 solvers. The architecture can be improved by using the solvers as agents that collaborate and communicate when solving ARC tasks. Such improvements would allow the solvers to support each other dynamically and achieve better performance on complex reasoning tasks.

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
