# OpenReview forum: "Reflection System for the Abstraction and Reasoning Corpus"
_AAAI.org/2025/Workshop/NeurMAD — AAAI 2025 Workshop NeurMAD Submission_

### Official Review · Reviewer_ssoG · 2024-12-24
**Interesting ideas, incremental improvements**

**Rating:** 7
**Confidence:** 5

**Review:**

Summary:
- This paper proposed an augmentation technique that improves performance for LLMs on ARC and shows that finetuning based on these augmentations further improves performance.
Additionally, it proposes a reflection system that enables the effective combination of multiple independent solvers.

Strength:
 - Identification of augmentations that bring improvement to ARC for LLMs
 - Evaluation of effect of QLoRA fine-tuning on AugARC for different models
 - In-depth analysis of solution overlap of different algorithms
 - Proposal of a reflection system that is able to select the most promising outputs among a selection of candidates
 - Running eval on complete evaluation set, enabling ease of comparison.
 - Good limitations section

Weaknesses:
 - Similarities of test-time augmentation scheme in AugARC to the one propozed by (Bober-Irizar and Banerjee 2024)
 - Minor improvements over DSL search using massively more compute (Claude 3 Opus is rumored to be quite large)
 - Exact prompting scheme for the reflection model is unclear
 - Missing insights into performance of reflection model
 - Ablations seem to be missing some scientific rigour


Questions and points for improvement:
- Figure 3: reflection model chooses the wrong solution in example 1, but the caption claimes it is the correct solution
- On page 4, you mention that for the remainder of the experiments, you are going to use AugARC. Since you call AugARC a benchmark, it is then rather confusing when you talk about ARC performance.
  It would make more sense to just call AugARC an augmentation for ARC, since it is not really a new benchmark but just an augmentation scheme, that can be used for training and at test time.
- Figure 4 would be easier to read if sorted by performance.
- Why is Figure 5a symmetric? Shouldn't it be normalized by the total number of tasks of the model on the bottom? With the current approach, you show overlap only in one direction.
- Also in Figure 5, you claim that the models are ordered by how much gain they add, but this does not seem to be the case?
- Which version of Gemini Pro are you using, I assume 1.5?
- You propose to use permutations as augmentations. If I've read the paper correctly, you never finetune on a dataset that includes permutations, which leaves it unclear whether permutations also yield a benefit for finetuning.
  A larger number of permutations could also lead to an imbalanced dataset due to tasks with different number of examples, eventually hampering the effectiveness of the augmentation.
- Did you also consider permuting the colors of the puzzles?
- It is commendable that you have evaluated many different models. However, your inconsistent use of different models for the different ablations and their inconsistent ordering in figures leaves a weird impression.
  * Ordering in Figures 4 and 5
  * Not all models of Figure 4 are present in Figure 5, how are they selected?
  * GPT models used as Reflection models, but not used as solvers
- Some more detailed ablations on the main contribution, the reflection system, would be interesting.
  * The upper bound of improvement over DSL search would be 23, the reflection system reaches 6.
  * In the setting where you use 3 solvers, does the fine-tunes Llama-3 8B even provide any potential new solved tasks to begin with?
  * Ablation with third solver for reflection system not really comparable to two solvers due to different reflection model used.
  * Can a reflection model provide a benefit if it is the same model that has been used as a solver, or does it need to be a different model?
- This is interesting work as it improves performance on ARC, but how does this contribute to the original goal of ARC, to achieve more intelligent and more human-like artificial systems?
- The conclusion does not fully reflect the content of the paper


The augmentations proposed in AugARC, mainly the permutations, could lead to an imbalanced dataset, eventually hampering the effectiveness of the augmentations.

Minor:
 - Weird paragraph break at the end of page 1
 - page 4: "Each solver solver independently and cannot..."
 - Weird paragraph break at the end of page 6
 - page 7: "by 6 ARC tasks. the Reflection System..."
 - There is also research on LLMs for the bloom model series (Camposampiero, Giacomo, et al. "Abstract visual reasoning enabled by language." Proceedings of the IEEE/CVF Conference on Computer Vision and Pattern Recognition. 2023.)
 - Further, (Wang, Ruocheng, et al. "Hypothesis search: Inductive reasoning with language models." arXiv preprint arXiv:2309.05660 (2023)) could also be included in the related work, as it is quite relevant.
  Unfortunately, they only provide results on a random subset of the ARC evaluation set, so direct comparison is not really possible.
 - ARC has been renamed to ARC-AGI

---

### Official Review · Reviewer_FLwR · 2024-12-29
**LLM with a program synthesis solver.**

**Rating:** 6
**Confidence:** 4

**Review:**

This work proposes a method combines ARC solutions from Large Language Models and a
Program Synthesis solver. Its performances are verified on the ARC and AugARC dataset.

As an ensemble method, it is not surprising to see better performances than methods that only use LLM or DSL solver. This work goes one step further by combing the 2 together to complement each other to achieve better performances. This is achieved by a reflection model.

Q: if the reflection model and the solver are the same LLM, will the reflection model prefers its own answer?

Agree with the authors that Claude 3 (sonnet) may have seen the ARC data and should be treated separately. However, this does not impact the final conclusion.

---

### Official Review · Reviewer_8jpk · 2024-12-30

**Rating:** 7
**Confidence:** 4

**Review:**

paper summary
The paper presents a novel approach to the Abstraction and Reasoning Corpus (ARC) that integrates LLMs with program synthesis solvers based on a DSL in a reflection-based architecture. It introduces AugARC, an enhanced benchmark to boost LLMs generalization.  This approach achieves a record 166/400 accuracy on ARC tasks, outperforming previous methods.

**Originality**
- **Strengths**:
The Reflection System leverages the strengths of both LLMs and program synthesis solvers based on a DSL, addressing the dataset limitations of the ARC through the AugARC benchmark. This benchmark enhances LLM generalization by incorporating augmented tasks, which broaden the scope of the original ARC. Furthermore, the system employs a self-reflection technique inspired by those used in LLMs, intelligently combining multiple solvers to tackle ARC tasks. The Reflection System demonstrates notable flexibility by supporting various solver types (e.g., LLMs and program synthesis tools) and allowing dynamic adjustment of solver configurations.

- **Weaknesses**:
The model lacks a theoretical analysis explaining why this reflection mechanism architecture improves task performance.

**Quality**
- **Strengths**:
The experimental setup is comprehensive and the performance improvement is significant.   Fine-tuning experiments with smaller LLMs (e.g., 7B and 13B parameters) demonstrate a significant improvement in ARC task performance, highlighting the value of data augmentation.

- **Weaknesses**:
 The theoretical foundation behind the reflection process is not fully explored. Specifically, how the reflection model determines the correct solution among solvers could benefit from more rigorous justification. The computational complexity of the reflection system, especially with multiple solvers and fine-tuned LLMs, is not deeply discussed. Most of the gains come from combining with Claude 3 Opus, raising questions about generalizability with other solvers.


**Significance**
- **Strengths**:
ARC is a challenging benchmark, and improving its performance meaningfully contributes to advancing AI’s ability for broad generalization and abstract reasoning. The system provides a new perspective on combining solvers, demonstrating the potential of reflection-based architectures for other reasoning benchmarks. The fine-tuning results suggest that even smaller LLMs can perform well on reasoning tasks, making the approach accessible for researchers without access to large-scale models.


- **Weaknesses**:
While the improvement over the ensemble system is clear, the gain of solving five additional tasks may not seem dramatic to practitioners.


**Questions and Suggestions for the Authors**
- Could the authors provide theoretical insights into why the reflection model effectively selects the correct solution?
- How does the computational complexity of the system (e.g., fine-tuning, running multiple solvers) scale with graph size or the number of tasks?
- While the results demonstrate state-of-the-art performance, could the authors include more extensive comparisons with other ensemble approaches, such as those combining different LLMs without program synthesis solvers?
- Since DSL Search contributes most solutions (160/400), is the reflection system’s performance reproducible with alternative program synthesis solvers?



**Limitations**
- The system relies on DSL Search and Claude 3 Opus for most of its performance gains.
- The computational cost of fine-tuning, reflection, and multi-solver integration could make the system infeasible for larger datasets or less computationally capable researchers.



**Ethics**

There are no obvious direct ethical concerns related to the method as it stands. The paper does not deal with sensitive data or produce sensitive content. The approach is a method improvement and not directly involved in human-facing decision-making applications at the evaluation stage. No unethical dataset or methodology usage is apparent. Thus, no ethical issues need to be flagged for special ethics review.

---

### Decision · Program_Chairs · 2024-12-30

**Decision:**

Reject

**Comment:**

This is a good paper. However, the method is still within the current paradigm of neural networks and does not fit the scope of this workshop.